# Transient Agarose Spot (TAS) Assay: A New Method to Investigate Cell Migration

**DOI:** 10.3390/ijms23042119

**Published:** 2022-02-14

**Authors:** Apor Veres-Székely, Domonkos Pap, Beáta Szebeni, László Őrfi, Csenge Szász, Csenge Pajtók, Eszter Lévai, Attila J. Szabó, Ádám Vannay

**Affiliations:** 11st Department of Pediatrics, Semmelweis University, 1083 Budapest, Hungary; pap.domonkos@med.semmelweis-univ.hu (D.P.); szebeni.beata@med.semmelweis-univ.hu (B.S.); szaszcsenge3@gmail.com (C.S.); pajtok.csenge@gmail.com (C.P.); levai.eszter@semmelweis-univ.hu (E.L.); szabo.attila@med.semmelweis-univ.hu (A.J.S.); vannay.adam@med.semmelweis-univ.hu (Á.V.); 2ELKH-SE Pediatrics and Nephrology Research Group, 1052 Budapest, Hungary; 3Department of Pharmaceutical Chemistry, Semmelweis University, 1092 Budapest, Hungary; lorfi@vichem.hu; 4Vichem Chemie Research Ltd., 1022 Budapest, Hungary

**Keywords:** fibroblast, migration, agarose, proliferation, collagen deposition, high throughput, assay

## Abstract

Fibroblasts play a central role in diseases associated with excessive deposition of extracellular matrix (ECM), including idiopathic pulmonary fibrosis. Investigation of different properties of fibroblasts, such as migration, proliferation, and collagen-rich ECM production is unavoidable both in basic research and in the development of antifibrotic drugs. In the present study we developed a cost-effective, 96-well plate-based method to examine the migration of fibroblasts, as an alternative approach to the gold standard scratch assay, which has numerous limitations. This article presents a detailed description of our transient agarose spot (TAS) assay, with instructions for its routine application. Advantages of combined use of different functional assays for fibroblast activation in drug development are also discussed by examining the effect of nintedanib—an FDA approved drug against IPF—on lung fibroblasts.

## 1. Introduction

Idiopathic pulmonary fibrosis (IPF) is an aggressive form of interstitial pneumonias characterized by excessive accumulation of fibroblasts and scar tissue formation, resulting in severe respiratory failure and high mortality rate [1,2]. Fibrosis independently from the etiology of the primary disease can virtually affect any organ, making this a key determinant of health [3]. Indeed, according to certain estimates tissue fibrosis is responsible for 45% of all death in the developed world [4].

Fibroblasts comprise the main cellular components of scar tissue. These cells can be described by their typical molecular biological and functional properties (Figure 1) [5,6] including enhanced migration due to their contractile stress fiber system, increased proliferation, and decreased apoptosis rate, resulting significant cell accumulation [7,8,9,10]. Activated fibroblasts are also responsible for the production of extracellular matrix (ECM), the structural element of the scar tissue, mainly composed of collagens and fibronectins [11,12]. These processes lead to excessive deposition of ECM which displaces native cells, leading finally to decreased organ function [13,14].

Nintedanib is one of the two approved drugs against IPF. Several preclinical and clinical studies have confirmed that multikinase inhibitors significantly slow progression of fibrosis by suppressing fibroblasts activation [15]. However, there is no approved pharmaceutical treatment available for other fibroproliferative diseases. Therefore, there is still a clear medical demand to develop antifibrotic drugs and targeting fibroblasts is a promising opportunity.

Several microplate-based assays have been described to investigate the different aspects of fibroblast activation. There are various well-constructed assays to investigate the proliferation (e.g., thiazolyl blue tetrazolium bromide—MTT; bromodeoxyuridine—BrdU) [16] or ECM production (e.g., SiriusRed staining) [17] of fibroblasts, however, the situation is different regarding their migration. The gold standard method is the scratch assay, which is based on graphical analysis of cell-free area created mechanically on a cell monolayer [18]. Despite its simplicity and cost-effectiveness, the scratch assay has significant limitations, including low reproducibility or high intra-assay variability [19,20,21,22].

In this article we describe our newly developed in vitro TAS assay, developed primarily for investigating fibroblast migration in a near high-throughput manner. Moreover, we also show how the set of microplate-based assays can be used to determine the main properties of fibroblasts, including migration, proliferation, and collagen deposition.

Beside the huge importance of fibroblast-related experiments, studying cell migration emerges also in other fields of research. Indeed, the adequate re-epithelization requires a proper balance in the migration and proliferation of epithelial cells during the repair process of injured lung, skin, or intestine [23,24,25]. In addition, the migration and proliferation of cancer cells determine the tumor progression and metastasis [26,27]. Therefore, in the present study we also examined the possible application of TAS assay for investigating the cellular mechanism of re-epithelization or tumor cell migration.

## 2. Results

### 2.1. Basic Settings of TAS Assay: Agarose Spot Stability and Optimal Cell Density

The stability of agarose gel spots was investigated over several consecutive days using MRC-5 lung fibroblasts. In ‘transient’ group, gel spots were removed 24 h after cell seeding, thereafter the rapid reduction in cell-free gap area was observed.

However, when the agarose gel spots were not removed (‘permanent’ group), the covered area remained cell-free for days, without any signs of under-agarose cell migration, allowing to change the medium and initiate the examination of migration at any time after optional pre-treatment steps (Figure 2).

The effect of cell confluence on the gap closure was also investigated using MRC-5 cells. The higher the cell count we used, the higher the confluence reached before gel removal, reaching the plateau phase at about 20,000 cell/well count (Figure 3a). Similarly, the increasing cell number resulted in accelerated gap closure, which was maximized at about 20,000 cell/well (Figure 3b). Based on the above-mentioned results, the confluency and the kinetics of gap closure showed a strongly positive correlation (Appendix A).

### 2.2. TAS Assay as Fibroblast Migration Assay

Cell migration of MRC-5 lung fibroblasts was investigated by TAS assay following various treatments. We found that addition of FBS into the culture media increased the rate of gap closure in a dose-dependent manner (Figure 4). Moreover, treatment with EGF also increased the extent of gap closure of MRC-5 cells (Figure 5).

### 2.3. Comparison of Different Gap Annotation Methods

To investigate the accuracy and reproducibility of the different methods, the size of the same gaps was determined by manual (Figure 6a) and automatic annotations (Figure 6b). In the case of brightfield or fluorescence images of MRC-5 cells, the manual or automatic annotation of the gap area showed a very high positive correlation (Figure 6c). While the brightfield images of NRK-49F cells were not suitable for the automatic annotation, because the software could not define the edge of the cell-free area, fluorescent images were appropriate for evaluation (Figure 6b).

Comparing the speed of the annotation methods, we found that although the manual annotation can be significantly accelerated by the use of a digitizer board or tablet, the automatic annotation of the images takes considerably fewer orders of magnitude of evaluation time (Figure 6d).

### 2.4. Comparison of Scratch and TAS Migration Assays

To compare their sensitivity and reproducibility, scratch and TAS migration assays were performed in parallel using MRC-5 cells.

Based on several independent experiments we found that while the confidence interval of initial gap size varied between 23 and 30% in the case of the scratch assay, it was only about 9% in the case of the TAS assay (Figure 7a). We found similar consistency while measuring directly the agarose spots without cell seeding (Appendix A). Beside the inconsistent size, the gap closure of scratched area is uneven and can only be documented by a series of images. In the case of TAS, the entire cell-free area can be investigated in one single field of view (Figure 7b).

The gap closure of MRC-5 cells showed similar kinetics comparing scratch (Figure 8a,c) and TAS (Figure 8b,d) migration assays. However, the intra-group variance determined by the coefficient of variation of group means was on average 3-fold higher in the scratch assay compared to the TAS assay, in the case of both relative (Figure 8a,b) and absolute (Figure 8c,d) gap size values.

### 2.5. Antifibrotic Effect of Nintedanib

Using a complex in vitro experimental setup, we investigated the effect of nintedanib on the main properties of MRC-5 lung fibroblasts including migration, viability, proliferation, and ECM production. To describe the migration ability of MRC-5 cells, the TAS assay was performed on EGF-treated fibroblasts in the absence or presence of nintedanib. We found that nintedanib significantly decreased the gap closure kinetics of both control (Figure 9a,b) and EGF-treated (Figure 9a,c) MRC-5 cells in a dose-dependent manner (Figure 9d).

At the end of the experiment cells were harvested and further analyzed by real-time PCR. Nintedanib decreased the mRNA expression of *PCNA* and *MKI67* in both control and EGF-treated MRC-5 cells (Figure 9f).

The proliferation of MRC-5 was investigated by MTT assay. Co-treatment with nintedanib decreased the PDGF-B-induced proliferation of fibroblasts in a dose-dependent manner (Figure 10a). Collagen deposition was determined by SiriusRed assay. Co-treatment with nintedanib decreased the TGF-ß induced collagen deposition of MRC-5 cells in a dose-dependent manner (Figure 10c).

Viability of cells was monitored by LDH cytotoxicity assay, performed on cell supernatants derived from TAS (Figure 9e), MTT (Figure 10b), and SiriusRed (Figure 10d) assays as well. We found no sign of cell death even at the highest dose of nintedanib during the experiments.

## 3. Discussion

IPF is one of the most aggressive forms of interstitial lung disease associated with high mortality rate and the lack of effective therapy. Although the etiology of IPF is unknown, the molecular and cellular mechanisms leading to tissue fibrosis are relatively well described revealing the crucial role of fibroblast activation [28]. Therefore, investigating the key features of fibroblasts, including their migratory capacity [29], proliferation [30], and ECM production [31], is inevitable to identify novel therapeutic targets and develop new antifibrotic compounds.

In the investigation of cell migration on two-dimensional monolayers, the scratch assay is the most commonly used method [32]. Although this assay has many advantages, including its simplicity or affordability, it also has serious limitations, which make its use as a high-throughput tool difficult [18]. Indeed, during the scratch assay, the cell-free gap is manually generated by scratching the surface of a confluent cell monolayer with a pipette tip on 6 or 12-well plates. Thereafter the migration of the cells and the kinetics of the gap closure is determined by graphical analysis. The main disadvantages of the scratch assay originate from the mechanical scratching itself, which causes damage of the cells, and also that of the plate surface, which has a significant impact on cell motility [33]. In addition, scratching with a pipette tip results in inconsistent initial gap sizes, which is reflected in the high intra- and inter-assay variability of the assay [32]. Therefore, alternative methods have been described to replace scratching recently. Among others, the cell-free zone can be created by heat stamp, laser, electricity, enzymatic digestion, or vacuum [20,34]. However, most of these methods have similar handicaps, including injured cells and surfaces, or resulting in gaps with irregular edges and sizes. From this point of view, a promising version could be the use of pre-installed physical barriers, where the cell-free zone is generated by plastic equipment or biocompatible gels [20,21,22]. Nevertheless, these commercially available kits are very expensive, and similarly to the scratch assay, work mostly with large surfaces (24 or 6-well plate) and thereby with large amounts of cells, reagents, and compounds, resulting in a low-throughput technique instead of a high-throughput one [20].

Therefore, in this study we aimed to develop a new assay eliminating the disadvantages of the above-mentioned techniques, using a simple, cost-effective, and high-throughput approach. In this method, which was termed as transient agarose spot (TAS) assay (Appendix A), liquid agarose hydrogel drops were placed in the middle of the wells on a 96-well plate to exclude the fibroblasts from a consistent, circular area (Figure 2). Until the hydrogel drops were removed by pipetting, they remained stable in the middle of the wells for several days despite manipulating the cells (e.g., medium change, serum starvation, or even performing transfection). Agarose is a widely used biocompatible gel with non-toxic, non-immunogenic properties [35,36]. However, after eliminating the hydrogel drop, fibroblasts started to migrate toward the cell-free surface of the gap, as revealed by the reduction in the initial size of gap areas (Figure 2).

In the first set of experiments, we investigated the impact of the cell density on the kinetics of gap closure in TAS assay. Not surprisingly, we found that cell number significantly influences the rate of gap closure (Figure 3). Based on our experiments we suggest the use of a nearly confluent (~90%) initial cell culture in order to receive sensitive measurement, but avoid formation of overlapping cell layers, which can cause inaccuracy during the experiments. The presence of BSA in the cell culture media further accelerates the gap closure of the investigated cells (Figure 4). Moreover, fibroblast migration can also be stimulated by adding different profibrotic growth factors [37,38,39], as demonstrated by our experiments on EGF-treated lung fibroblasts (Figure 5). In summary, varying with the initial cell count, the amount of BSA, and other optional stimulants, our TAS assay can be easily fine-tuned to examine the effect of various factors on cell migration.

Although in this study we focused on lung fibroblasts, the TAS assay can be performed on other fibroblasts (Appendix A) or even other cell types, as well. For example, using epithelial cells (Appendix A), the mechanism of reepithelization and wound healing can be properly investigated. In the case of cancer cells (Appendix A), TAS can be interpreted as an useful assay of invasion, which is a determining process in tumor metastasis [37].

During the data evaluation of the TAS assay, the cell migration is determined after taking serial photographs of the cell-free area at certain intervals. Thereafter, the gaps can be easily assigned using graphical software (e.g., ImageJ) without any previous experience, and then the size of the gap area can be determined using a single measurement command. Nevertheless, this technique is quite time-consuming and monotonous, therefore, we investigated whether manual annotation using a standard computer mouse can be accelerated by a digitizer board (without display) or tablet (with display), or can be replaced by automatic assignment (Figure 6), using macros (described in the methods section). The automatic annotation resulted in significantly shorter evaluation time per image (Figure 6d), however, optimal macro-options and setups should be previously adjusted in order to enable adequate recognition of gap edges. The automatic analysis proved to be well-applicable in the case of many cell types, including MRC-5 lung fibroblasts, but there are some cells, such as the more flattened NRK-49F renal fibroblasts, with their brightfield images showing lower contrast, causing difficulties in the software-driven automatic annotation of the gaps. To avoid this problem, the flattened cells can be labeled with various non-toxic stains. In our experiments, DiI membrane-dye was used to generate a stable fluorescence signal making it possible to take images with adequate contrast (Figure 6a). In this case, after a small modification in the macros, the automatic annotation became feasible with perfect correlation compared to manual ones (Figure 6b,c).

After setting up the TAS assay, we compared its laboratory use and sensitivity with that of the gold standard scratch assay on EGF-stimulated MRC-5 lung fibroblasts (Figure 7 and Figure 8). The benefits of the 96-well plate-based TAS assay, beside the requirement of fewer raw materials, were immediately conspicuous. Indeed, in the case of TAS the whole gap area fits into a single field-of view of the microscope, which makes the easy automated documentation of the entire gap possible, reducing the need for human resources (Figure 7b). On the contrary, in the case of the scratch assay the process is more complex and the identification of the same area at the different investigated time points is a real challenge [20].

In accordance with the literature, the standard deviation of the initial gap area was about 3-fold higher in the case of the scratch compared to the TAS assay (Figure 7a) [37]. This is a serious limitation since the higher standard deviation of the initial gap area may later result in an even higher standard deviation of the remaining cell-free gap area. Indeed, in line with this consideration, we found that although the kinetics of gap closure is similar in both assays, the TAS assay seemed to have a significantly higher resolution, revealed by the 3-fold smaller deviation of the corresponding gap closure values compared to the scratch assay (Figure 8). To avoid the distortion due to the inconsistent initial gap size, which can appear in case of relative percentage values normalized to the cell-free area at 0 h (Figure 8a,b), the gap closure was also determined by absolute, pixel^2^ values (Figure 8c,d). Nevertheless, we found the same difference in the intra-assay variability of the two methods in favor of the TAS assay.

The high accuracy of the TAS assay allows to detect the migration differences even in the early phase of the experiment, such as 3 h after the onset of treatments (Figure 5b). In summary, our results confirmed that the TAS assay is an improved alternative to the scratch assay, retaining its advantages and eliminating most of its limitations (Table 1). However, it should be noted that gap closure-based migration assays, including scratch or TAS assays are dedicated to investigating the collective cell migration, rather than individual single-cell motility. However, this type of cell migration is a hallmark of the tissue remodeling processes, including fibrosis, reepithelization, wound healing, and cancer invasion [40,41]. At this point it had to be noted that, despite their similar elements, the TAS assay should not be confused with the under-agarose method. In the TAS assay the agarose spot represents a mold, which excludes the cells from a certain area during their attachment to the surface of the plate. In contrast, during the under-agarose assay chemotaxis is induced by chemoattractant molecules dissolved in agarose [42,43,44]. Via the latter method, the individual movement of sperm cells or leukocytes can be modeled and quantified by determining the number of cells that migrated under the permanent agarose.

Finally, we demonstrated the usability of the combination of the TAS migration assay with other assays to investigate the main properties of activated fibroblasts in vitro. In these experiments, although there are commercially available kits, we used our self-optimized, 96-well plate based, cost-effective in vitro assays to determine cell migration, proliferation, and collagen deposition (for more detailed information please see the corresponding section of the methods).

First, we demonstrated that nintedanib, one of the approved drugs against lung fibrosis in IPF, decreases the migration of fibroblast in a dose-dependent manner (Figure 9a–d). As fibroblast migration can be continuously monitored during the TAS assay, even at short, hourly intervals, the experiment can be terminated at any time when significant migration is detected and both the cells, and also their supernatants, can be further analyzed. Indeed, in our present study, at the end of the TAS assay a decreasing effect of nintedanib on the expression of cell cycle regulators was demonstrated by PCR analysis (Figure 9f), revealing its effect on fibroblast proliferation and accumulation [45,46,47]. We confirmed the antiproliferative effect of nintedanib independently using an MTT assay on PDGF-B-stimulated cells as well (Figure 10a).

Moreover, the SiriusRed assay was also performed to determine the extent of collagen deposition of fibroblasts after their activation with TGF-ß. In this experiment we demonstrated that nintedanib decreases the ECM production of the fibroblasts (Figure 10c).

In addition to the previously described experiments, investigating the activation of fibroblasts, LDH assay was performed to gain further information. Since LDH is an intracellular enzyme, its presence in the supernatant of the cells suggests that the experimental conditions were toxic for the cells, which can significantly influence the result of the discussed assays.

Therefore, during our experiments we measured the LDH activity in the cellular supernatants derived from TAS (Figure 9e), MTT (Figure 10b) and also from SiriusRed (Figure 10d) assays. Thereby, we demonstrated that the observed inhibitory effects of nintedanib were not due to its cytotoxic effect but to its specific effects on fibroblast activation.

Taken together, these simple and cost-effective assays, including TAS, MTT, SiriusRed, and LDH are complementary and have the ability to describe the different aspects of fibroblast activation, including migration, proliferation, and ECM production of the main effector cells of fibrosis.

## 4. Materials and Methods

### 4.1. Cell Lines

MRC-5 (#CCL-171) human lung fibroblast, NRK-49F (#CRL-1570) rat kidney fibroblast, A549 (#CRM-CCL-185) human lung epithelial cell, and Caco-2 (#HTB-37) human colon carcinoma cell lines (American Type Culture Collection (ATCC), Manassas, VA, USA) were cultured in Dulbecco’s Modified Eagle Medium (Thermo Fisher Scientific, Waltham, MA, USA), HT-29 (#HTB-38) human colon carcinoma cell line (ATCC) was cultured in McCoy’s 5A Medium (Thermo Fisher Scientific) supplemented with 10% heat-inactivated fetal bovine serum (FBS) (Invitrogen, Waltham, MA, USA) and 1% penicillin and streptomycin (Merck, Kenilworth, NJ, USA) mixture under standard cell culture conditions (37 °C, humidified, 5% CO_2_). During in vitro experiments, recombinant epidermal growth factor (EGF, 10 ng/mL, #236-EG, R&D Systems, Minneapolis, MN, USA), recombinant platelet-derived growth factor B (PDGF-B, 10 ng/mL, #520-BB, R&D Systems), recombinant transforming growth factor beta 1 (TGB-ß, 1 nM, #PHG9204, Thermo Fisher Scientific), and nintedanib (Nint, 0.01–10 μM, Vichem Chemie Research, Budapest, Hungary) were used. Control cells were treated only with the corresponding solvents (EGF, PDGF-B: phosphate buffered saline (PBS), Nint: DMSO, TGF-β: 4 mM HCl) in equal volumes.

### 4.2. Transient Agarose Spot (TAS) Migration Assay

To perform TAS assay, 2 μL of hot 0.1% agarose (Merck) solution (in sterile H_2_O) was placed in the middle of each well of a 96-well tissue culture plate (Sarstedt, Newton, MA, USA), then gel droplets were allowed to polymerize for 15 min at room temperature. Thereafter, cells were seeded (*n* = 5–8 well/treatment group) at a density to reach near full confluence (2 × 10^4^ cells/well (otherwise indicated) MRC-5, 10^4^ cells/well NRK-49F, 2 × 10^4^ cells/well A549, 6 × 10^4^ cells/well Caco-2, 4 × 10^4^ cells/well HT-29). After 24 h of plating, media was removed from cells and the agarose spots were gently aspirated by a 100 μL pipette from above, without touching the cell monolayer (further instruction can be found in Appendix A and in the section ‘Tips and tricks’). To remove debris and unattached cells, wells were washed three times with 200 μL sterile PBS. Then, cells were treated with recombinant cytokines and/or compounds diluted in 100 μL culture medium containing 0.1% FBS, unless otherwise indicated. In certain experiments, as an additional part of TAS assay, MRC-5 cells were incubated in 0.1 mg/mL DiI solution (#D282, Thermo Fisher Scientific) for 2 h before the agarose removal and treatments. The main steps of TAS assay are illustrated in Appendix A.

### 4.3. Scratch Assay

To perform scratch assay [18], MRC-5 cells were seeded (*n* = 6 well/treatment group) into 12-well tissue culture plates (Sarstedt) at a density of 4 × 10^5^ cells/well. After 24 h of plating, media were aspired from cells and the monolayer was scratched with a single decisive movement using a 200 μL pipette tip. To remove debris and unattached cells, wells were washed three times with 2 mL sterile PBS. Then, cells were treated with recombinant EGF diluted in 1 mL culture medium containing 0.1% FBS.

### 4.4. Data Acquisition

Brightfield or fluorescence images of each well were taken using an Olympus IX81 microscope system (Olympus Corporation, Tokyo, Japan) at various time points after the treatments. Cell-free gap areas were annotated manually by standard computer mouse, digitizer board without display or tablet or automatically and measured using ImageJ 1.48v software (National Institutes of Health, Bethesda, Rockville, MD, USA), finally determined as a ratio (%) of initial gap area at 0 h:(1)$$\mathrm{gap}\;\mathrm{area}\;[\%]=\frac{\mathrm{actual}\;\mathrm{gap}\;\mathrm{area}}{\mathrm{initial}\;\mathrm{gap}\;\mathrm{area}}\;\times \;100$$

In some cases, the gap closure was described by the absolute value decrease in cell-free area:(2)$$\Delta \mathrm{gap}\;\mathrm{area}\;\left[{\mathrm{pixel}}^{2}\right]=\mathrm{initial}\;\mathrm{gap}\;\mathrm{area}-\mathrm{actual}\;\mathrm{gap}\;\mathrm{area}$$

During automatic data analysis, the following macros were used. ***Highlighted parameters*** were set up and verified in each individual experiment.

Brightfield images:

run(“Find Edges”);

setAutoThreshold(“Default”);

//run(“Threshold...”);

setThreshold(***0, 20***);

run(“Analyze Particles...”, “size=***2,000,000***-Infinity show=Outlines summarize”);

close();

Fluorescence images:

run(“RGB Stack”);

run(“Next Slice [>]”);

run(“Delete Slice”);

run(“Next Slice [>]”);

run(“Delete Slice”);

run(“Find Edges”, “slice”);

setAutoThreshold(“Default”);

//run(“Threshold...”);

setThreshold(***0, 20***);

run(“Analyze Particles...”, “size=***2,000,000***-Infinity show=Outlines display clear summarize slice”);

close();

### 4.5. Cell Confluency

To determine the cell confluency, cells were labeled with DiI as described above in the section ‘Transient agarose spot (TAS) migration assay’. Thereafter fluorescent images were taken with wide field of view and analyzed using ImageJ software. During the data acquisition, the extent of black pixels was measured using ‘color threshold’ and ‘measure’ commands, the gel spot area was determined by annotation, and the confluency was calculated based on the following equation:(3)$$\mathrm{confluency}\;[\%]=\left(1-\frac{\mathrm{black}\;\mathrm{pixel}\;\mathrm{area}-\mathrm{gel}\;\mathrm{spot}\;\mathrm{area}}{\mathrm{total}\;\mathrm{area}-\mathrm{gel}\;\mathrm{spot}\;\mathrm{area}}\right)\;\times \;100$$

### 4.6. LDH Cytotoxicity Assay

The extent of cell death was determined by a colorimetric method, based on the lactate dehydrogenase (LDH) enzyme activity in the supernatant, released from damaged cells [48]. Equal volumes (40 μL) of aspired media were mixed in a sterile 96-well plate with LDH reagent, containing 109 mM lactate, 3.3 mM ß-nicotinamide-adenine-dinucleotide-hydrate (#N7004), 2.2 U/mL diaphorase (#D2197), 3 mM TRIS, 30 mM HEPES, 10 mM NaCl, 350 μM thiazolyl blue tetrazolium bromide (all reagents were purchased from Merck), then incubated at 37 °C for 1 h. Absorbance was recorded at 570 nm and at 690 nm as background in a SPECTROstar Nano microplate reader using SPECTROstar Nano MARS v3.32 software (BMG Labtech, Ortenberg, Germany).

### 4.7. MTT Cell Proliferation Assay

To perform MTT assay, MRC-5 cells were seeded (*n* = 6 well/treatment group) into 96-well tissue culture plates at a density of 4 × 10^3^ cells/well. After 24 h of plating, cells were treated for 24 h with recombinant PDGF-B in the absence or presence of nintedanib diluted in culture medium containing 0.1% FBS.

The rate of cell proliferation was determined by a colorimetric method, based on the intracellular mitochondrial dehydrogenase activity of the attached cells [49]. Then, 10 μL of MTT reagent, containing 5 mg/mL thiazolyl blue tetrazolium bromide (diluted in sterile H_2_O) was added into each well including cells and 100 μL of supernatant as well, then incubated at 37 °C for 4 h. Thereafter, the supernatants were removed from cells using a pipette, and the intracellular MTT crystals were dissolved by adding 100 μL 1:1 mixture of DMSO and ethanol (all reagents were purchased from Merck). Absorbance was recorded at 570 nm and at 690 nm as background in a SPECTROstar Nano microplate reader using SPECTROstar Nano MARS v3.32 software.

### 4.8. SiriusRed Collagen Detection Assay

To perform SiriusRed assay, MRC-5 cells were seeded (*n* = 6 well/treatment group) into 96-well tissue culture plates at a density of 10^4^ cells/well. After 24 h of plating, cells were treated for 48 h with recombinant TGF-ß in the absence or presence of nintedanib diluted in culture medium containing 0.1% FBS and 100 μM ascorbate (Merck).

The collagen deposition was determined based on a basic histological dye SiriusRed, incorporating into the triple helical collagen molecules [50]. After removing supernatants, cells were incubated in a fixative solution containing 26% EtOH, 3.7% formaldehyde, 2% glacial acetic acid for 15 min at room temperature. Samples were stained for 1 h at room temperature with 0.1% solution of SiriusRed (DirectRed80) dissolved in 1% acetic acid, then washed three times with 200 μL of 0.1 M HCl, and finally the bounded dye was dissolved by adding 100 μL of 0.1 M NaOH (all reagents were purchased from Merck). Absorbance was recorded at 544 nm and at 690 nm as background in a SPECTROstar Nano microplate reader using SPECTROstar Nano MARS v3.32 software.

### 4.9. RNA Isolation and cDNA Synthesis

Total RNA was isolated from MRC-5 cells by Geneaid Total RNA Mini Kit (Geneaid Biotech, New Taipei City, Taiwan). Equal RNA was reverse-transcribed using Maxima First Strand cDNA Synthesis Kit for RT-qPCR (Thermo Fisher Scientific) to generate first-stranded cDNA. The mRNA expressions were determined by real-time PCR using LightCycler 480 SYBR Green I Master enzyme mix on a LightCycler 96 system (Roche Diagnostics, Indianapolis, IN, USA). PCR primers (Table 2) were designed as previously described [51,52] and synthetized by Integrated DNA Technologies (IDT, Coralville, IA, USA). Results were analyzed using LightCycler 96 software v1.1.0.1320 (Roche Diagnostics). Relative mRNA expression was determined by comparison with *RN18S* as internal control using the ∆∆Ct method [53]. Data were normalized and presented as the ratio of their control group values.

### 4.10. Statistical Analysis

Statistical evaluation of data was performed using GraphPad Prism 8.01 software (GraphPad Software Inc., San Diego, CA, USA) using an ordinary one-way or two-way ANOVAs with Dunnett’s tests for multiple comparisons and Pearson’s correlation for correlation analyses. *p* ≤ 0.05 was considered as statistically significant. Unless otherwise indicated, results are illustrated as mean ± SD of the corresponding treatment groups. The applied tests, significances, and number of elements (n) are indicated in each figure legend.

## 5. Conclusions

In summary, in the present study we presented a new 96-well plate based TAS assay, developed to investigate fibroblast migration. We also demonstrated the combined use of a set of in vitro assays, which can be used in fibrosis-related basic research, and even in high-throughput drug screenings. These functional assays together widely describe the main properties of fibroblasts, including migration, proliferation, viability, and collagen deposition, thereby contributing to an increasingly urgent understanding of the scarring process.

## 6. Tips and Tricks for TAS Assay

Working with hot fluids may cause pipetting inaccuracies → after boiling, agarose solution should be left for about 10 min to cool down.Electrostatic charging may cause difficulties in agarose dropping → static electricity should be eliminated by a grounded device or ionizing bar.Drop size can be decreased in order to achieve more spectacular relative values ←→ however, 2 μL droplets are optimal for microscopy using a 4× objective, and the smaller the volume is, the harder it is to ensure pipetting accuracy.The pipetting of agarose spots can be accelerated by using multichannel pipettes.Drying-up of agarose spots results in improper cell-free area-making → plates should be kept covered during gel polymerization phase.Some cell types (e.g., large, flat fibroblasts) may attach in a shorter amount of time → the cell seeding step can be shortened to 3–4 h instead of overnight.Some cell types (e.g., small, rounded carcinoma cells) may attach less strongly → the washing step should be gentle but thorough to eliminate floating cells, which could form colonies in the cell-free area.During the removal of agarose spots, the pipette tip should be carefully approached to the top of the gel. At the moment of contact, the refraction of the gel will change, and it can be sucked by a single movement.

## Figures and Tables

**Figure 1 ijms-23-02119-f001:**
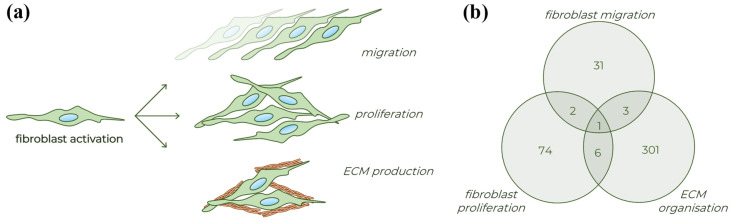
Different processes of fibroblast activation. (**a**) Schematic figure about the typical properties of activated fibroblasts. (**b**) Venn diagram presenting the number of human genes involved in the various fibroblast related biological processes based on the Gene Ontology database (detailed in Appendix A).

**Figure 2 ijms-23-02119-f002:**
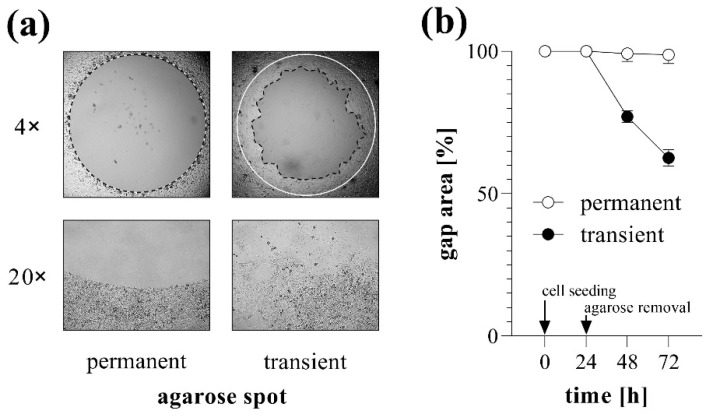
Gap closure in case of permanent and transient agarose spots. To investigate the stability of agarose gel spots, TAS migration assay was performed on MRC-5 cells. (**a**) Cell-free zone areas were analyzed graphically after brightfield microscopy. Lines in representative images indicate the gap edges at 0 (white) and 48 (black) hours after gel removal. Pictures taken with 20× objectives show the edges of cell-free zones. (**b**) The gap closure was monitored for 72 h after cell seeding. Results are presented as mean ± SD (*n* = 6).

**Figure 3 ijms-23-02119-f003:**
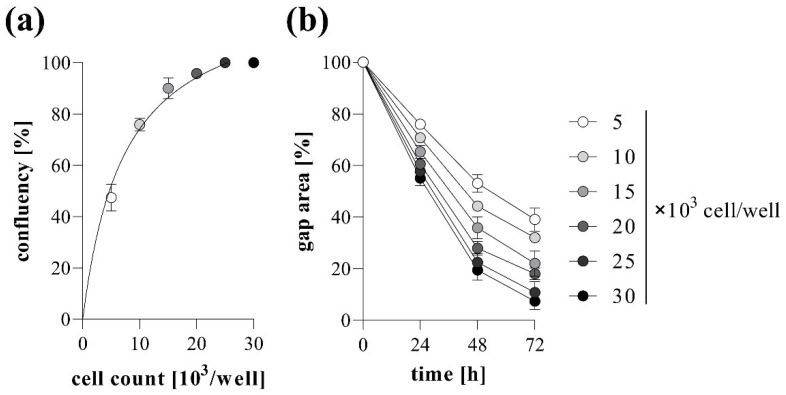
Effect of cell density on gap closure. To determine optimal cell count, TAS migration assay was performed on MRC-5 cells. (**a**) The resulting confluencies and the (**b**) kinetics of gap closure in the case of various cell counts were determined graphically. Results are presented as mean ± SD (*n* = 5).

**Figure 4 ijms-23-02119-f004:**
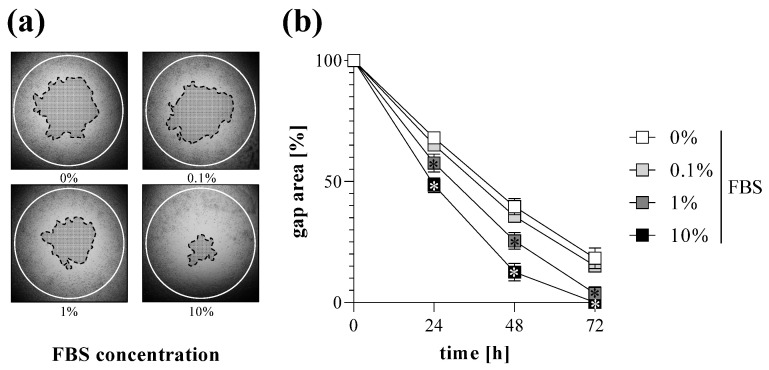
Effect of FBS treatment on gap closure. To determine the effect of serum addition, TAS migration assay was performed on MRC-5 cells. (**a**) Cell-free zone areas were analyzed graphically after brightfield microscopy. Lines in representative images indicate the gap edges at 0 (white) and 48 (black) hours after gel removal. (**b**) The gap closure was monitored for 72 h after gel removal. Results are presented as mean ± SD (*n* = 8). * *p* < 0.05 vs. 0% FBS at the concerning time (two-way ANOVA).

**Figure 5 ijms-23-02119-f005:**
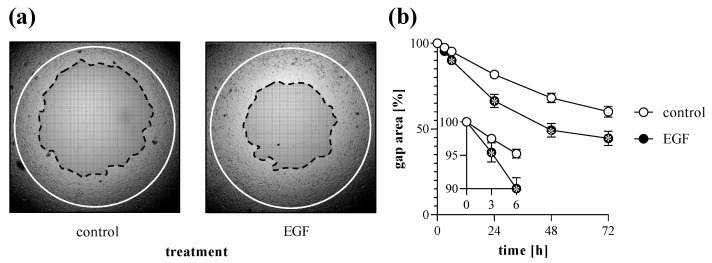
Effect of growth factor treatment on gap closure. To determine the effect of EGF treatment, TAS migration assay was performed on MRC-5 cells. (**a**) Cell-free zone areas were analyzed graphically after brightfield microscopy. Lines in representative images indicate the gap edges at 0 (white) and 48 (black) hours after gel removal. (**b**) The gap closure was monitored for 72 h after gel removal. Results are presented as mean ± SD (*n* = 8). * *p* < 0.05 vs. 0% control at the concerning time (two-way ANOVA).

**Figure 6 ijms-23-02119-f006:**
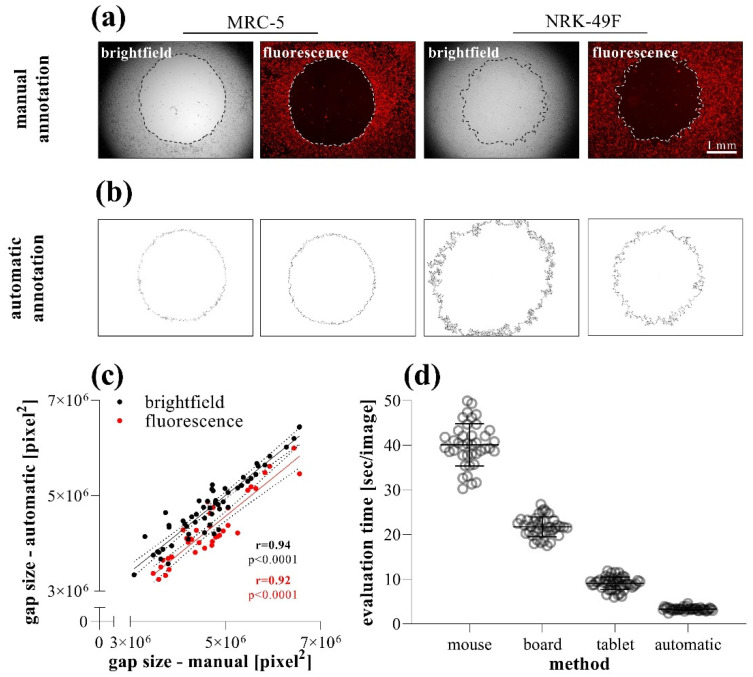
Comparison of data evaluation methods in TAS migration assay. (**a**) Brightfield and (**b**) fluorescence images were taken of DiI stained MRC-5 and NRK-49F cells. The same images (*n* = 40) of MRC-5 cells were analyzed after manual and automatic annotation, then the (**c**) resulting gap areas and the (**d**) required evaluation times were compared. Correlation was determined by Pearson’s coefficients (r).

**Figure 7 ijms-23-02119-f007:**
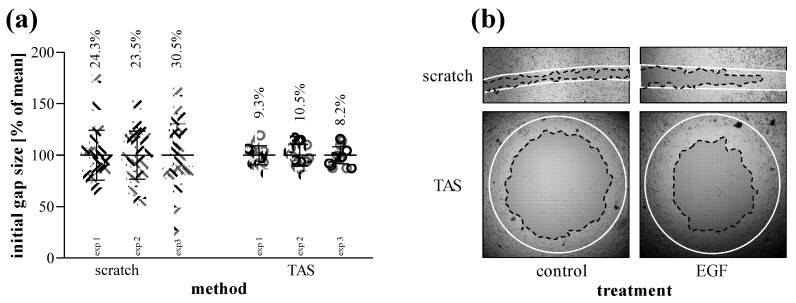
Reproducibility of scratch and TAS migration assays. (**a**) Consistency of initial gap sizes was determined in independent experiments performed on MRC-5 cells (*n* = 24–30 in each 3-3 experiments). Percentage values indicate the coefficient of variation of the concerning groups. (**b**) Evenness of gap closure was investigated on control and EGF-treated cells, as demonstrated in representative microscopic images. Lines in representative images indicate the gap edges at 0 (white) and 48 (black) hours after gel removal.

**Figure 8 ijms-23-02119-f008:**
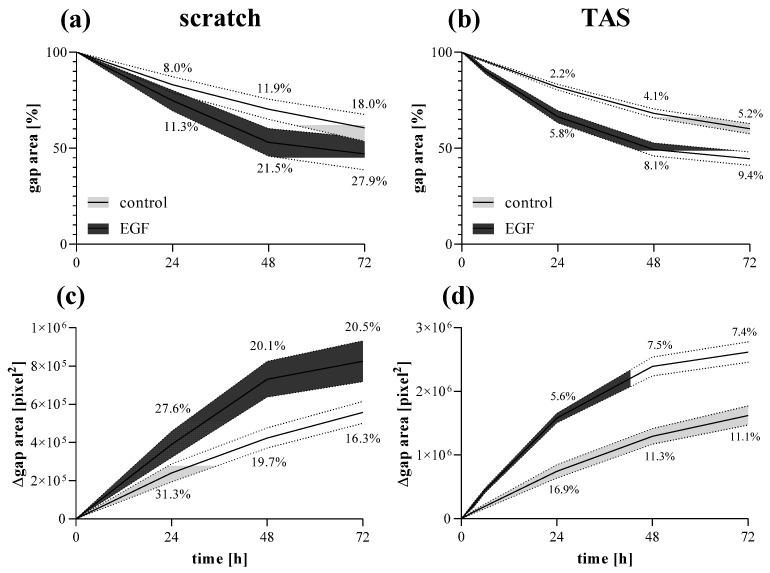
Fibroblast migration determined by different methods. Gap closure kinetics of control and EGF-activated MRC-5 cells were investigated by (**a**,**c**) scratch and (**b**,**d**) TAS migration assays. Alteration in gap size was determined by (**a**,**b**) relative (percentage of initial size) or (**c**,**d**) absolute (Δpixel^2^) values. Results are presented as mean ± deviation, where line widths and percentage values indicate the coefficient of variation of the concerning groups (*n* = 6).

**Figure 9 ijms-23-02119-f009:**
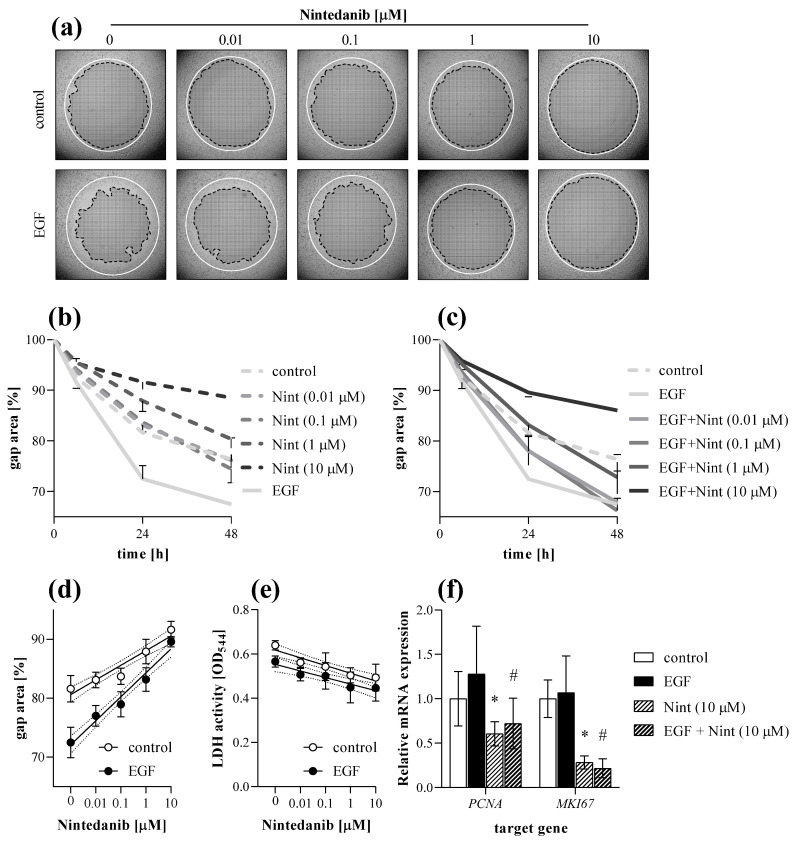
Effect of nintedanib on fibroblast migration. The gap closure kinetics of (**a**,**b**) control and (**a**,**c**) EGF-activated MRC-5 cells were determined by TAS assay in the absence or presence of nintedanib (Nint). Lines in section (**a**) indicate the gap edges at 0 (white) and 48 (black) hours after gel removal. (**d**) Dose-dependence was investigated on data derived from 24 h after gel removal. (**e**) To monitor cytotoxicity, LDH assay was performed on cell supernatants. (**f**) Relative mRNA expressions were determined by comparison with *RN18S* ribosomal RNA as internal control and normalized as the ratio of the control group. Results are presented as mean ± SD (*n* = 6). * *p* < 0.05 ‘Nint (10 μM)’ vs. ‘control’, # *p* < 0.05 ‘EGF + Nint (10 μM)’ vs. ‘EGF’ (two-way ANOVA).

**Figure 10 ijms-23-02119-f010:**
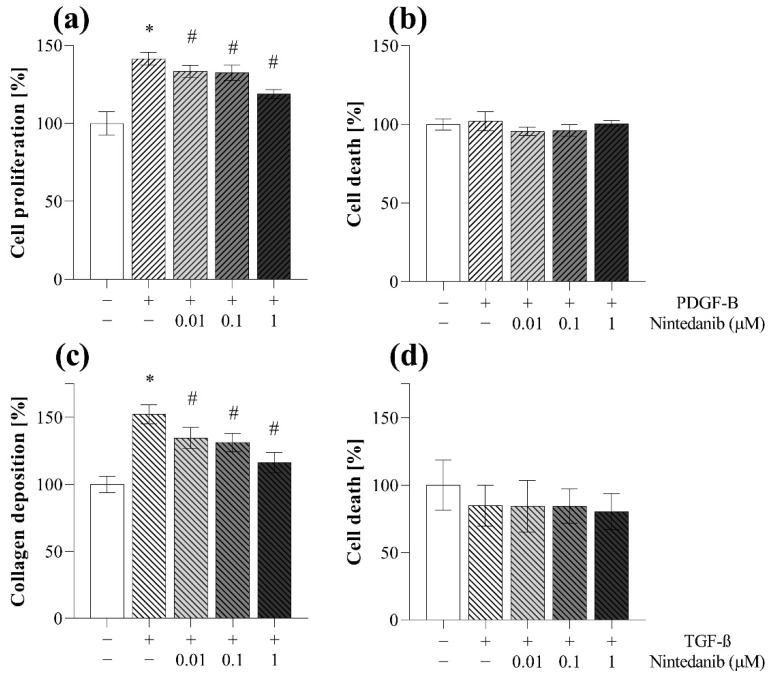
Effect of nintedanib on fibroblast proliferation and collagen deposition. (**a**,**b**) PDGF-B-induced cell proliferation and (**c**,**d**) TGF-ß-induced collagen deposition of MRC-5 cells in the absence or presence of nintedanib was investigated by (**a**) MTT and (**c**) SiriusRed assays. (**b**,**d**) To monitor cytotoxicity, LDH analysis was performed on cell supernatants. Results are presented as mean ± SD (*n* = 6). * *p* < 0.05 vs. control, # *p* < 0.05 vs. TGF-ß.

**Table 1 ijms-23-02119-t001:** Comparison of the advantages (+) and limitations (−) of scratch and TAS migration assays.

Assay Characteristics	Scratch Assay	TAS Assay
Cost-effective	+	+
Minimal equipment	+	+
Easy to perform	+	+
Consistent gap size	−	+
Well reproducible	−	+
Easy gap relocation during repetitive imaging	−	+
Intact cells and plate surface	−	+
Small surface and volumes (96-well plate based)	−	+
Low intra- and inter-assay deviation	−	+
Automatable	−	+

**Table 2 ijms-23-02119-t002:** Nucleotide sequences of primer pairs applied for the real-time PCR detection.

Gene	NCBI Ref. Seq.	Primer Pairs
*MKI67*	NM_002417.4	F: 5′-CCC CTA CGG ATT ATA CTC AAC TTA-3′ R: 5′-TGT AAT ATT GCC TCC TGC TCA T-3′
*PCNA*	NM_002592.2	F: 5′-GCG GTC TGA GGG CTT CGA CAC CTA-3′ R: 5′-CCG CGT TAT CTT CGG CCC TTA GTG-3′
*RN18S*	HQ387008.1	F: 5′-GGC GGC GAC GAC CCA TTC-3′ R: 5′-TGG ATG TGG TAG CCG TTT CTC AGG-3′

Abbreviations: ref. seq.: reference sequence; F: forward; R: reverse.

## Data Availability

Data is contained within the article or Appendix A.

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
