# Peer review of "Transient Agarose Spot (TAS) Assay: A New Method to Investigate Cell Migration"

_ijms, 2022, doi:10.3390/ijms23042119_

Round 1
Reviewer 1 Report
The manuscript is written in good English and contains high-quality figures, which clearly illustrate the ideas, design of experiments, and results. The authors developed an original method of examination of cell migration, which is described very well and can be useful for many researchers. The results are presented pretty clearly and they strictly support the conclusion.
I have only one minor comment to the authors: it would be great to slightly extend the introduction to make it clearer for readers from different research areas.
Author Response
Thank You for the careful reviewing of the manuscript and also for Your suggestion, accordingly we supplemented the introduction with new information so that the purpose and the use of the developed TAS assay be clear for the professionals of the different scientific area. Proofreading was applied for the changes in the manuscript (see lines 61-68).

Reviewer 2 Report
This article presents a method to study cell migration using fibroblasts in combination with fibroblast activation using FDA approved drug nintedanib. Transient Agarose Spot (TAS) assay was used to study MRC-5 cells. Bright-field microscopy was used to study permanent and transient spots for 72 hours. The authors studied the gap closure wrt cell seeding density at 0 hours. TAS assay was also performed under different FBS concentrations of the culture medium. The effect of EGF treatment on MRC-5 cells was also performed via TAS assay. A comparison has been made between manual annotation and automatic annotation for bright-field and fluorescent imaging. Another comparison between scratch and TAS assays has been performed, and the TAS assay is better in terms of statistics. Finally, the effect of nintedanib has been assayed in a dose-dependent manner. The whole experiments are designed and conducted systematically. The results are presented with good quality scientific figures. Therefore, I recommend the article be published in this journal.
Author Response
Thank You for the reviewing and also for the positive assessment of our study.
Reviewer 3 Report
In the manuscript "Transient Agarose Spot (TAS) assay: a new method to investigate cell migration", Veres-Szekely et al. present a high-throughput method to quantify collective cell migration. While the manuscript is well-presented, and the method is well-validated, proving better than the commonly used scratch assays, I find that this manuscript does not fall within the scope of the IJMS as it does not qualify as a "breakthrough experimental technical progress of broad interest in biology, chemistry and medicine". I would advise the authors to submit the manuscript to a protocol-focused journal, such as STAR Protocols or JoVE.
Author Response
We are glad that the quality, the validation and also the presentation of our study won your approval, however we must to argue with your comment regarding the positioning of the manuscript in other journals.
The aims of the special issue (Molecular Pathology of Idiopathic Pulmonary Fibrosis 2.0) is to provide answers to the unresolved questions on the Molecular Pathology of Idiopathic Pulmonary Fibrosis.
Fibroblast activation, which is a hallmark of the pathomechanism of IPF, is characterised by increased proliferation, migration and ECM production of fibroblasts. All these processes have distinct molecular biological background as it is mentioned in the section of Introduction (please see lines 31-38 and also Figure 1).
To study the complex mechanism of fibrosis all these processes, including migration of fibroblast should be investigated.
It is also important to mention that the process of tissue fibrosis is similar regarding the different diseases and organs, therefore understanding the molecular processes of IPF may also contribute to the understanding and treatment of other fibroproliferative diseases. The importance of fibrosis related studies is underlined by the estimation suggesting that almost every second death is related to tissue fibrosis.
Although some of tests have already been used to examine the migration of fibroblasts, they are complex and difficult to reproduce. In contrast, TAS assay is fast, unexpensive, well reproducible and need no special equipment to perform it. All these features of TAS assay facilitate its widespread use to investigate fibroblast migration and thus the better understanding of this pathomechanical segment of fibrosis.
Although, there are FDA approved drug to treat IPF, including nintedanib and pirferidone, their use did not lead to a real breakthrough in the treatment of the patients. Moreover, there is no approved drug at all against other fibroproliferative diseases.
Therefore, we believe that it is important that the scientific community be aware of those methods which facilitate the understanding of the complex mechanism of IPF and tissue scarring, as well as to have the right assay to test the effect of new antifibrotic compounds.

Reviewer 4 Report
Manuscript No. ijms-1542157
„Transient Agarose Spot (TAS) assay: a new method to investigate cell migration” for International Journal of Molecular Sciences
Comments:
- Materials and methods. 4.1. Please provide ATCC catalog number of the cell line.
- Results. How was gel removed from the microplate wells? I am asking the authors to include a short explanation in the text. The authors mentioned pipetting the gel. Doesn't this carry potential cell damage? Moreover, how confident are the authors that the cells will not grow under the agarose?
- How can I be sure that the agarose drops are comparable? Was it only stated to be equal gel volume or was it otherwise standardized?
- I am asking the authors to compare if, instead of the gel, the insertion and removal of the plastic mold would not have a similar effect? I understand this has been mentioned in the discussion, but micro inserts for 96-well plates can be used as well.
- I am asking the authors to briefly explain the obtained results in the discussion and not to duplicate the information from the Results section. I am referring to the second part of the discussion regarding the nintedanib effect.
Reviewer 5 Report
Veres-Székely et al present a laboratory technique for in vitro assessment of fibroblast migration. Their work is quite detailed and appears to be a useful presentation of experimental data. The style needs attention to proper punctuation and improved English usage, but otherwise meets expectations of an original contribution.
Major points
The authors may wish to back off any claim to priority concerning ‘agarose spot’ reagents to track chemotactic response driving cell migration. This is because Ahmed et al (Sci Rep 2017 Apr 21;7(1):1075) already published on this topic. Unfortunately, that work is missing from the reference list. Thus, these investigators need to explain how their ‘new method’ differs from previous research.
Minor issues
The scope of this project is limited to fibroblasts. The authors should consider changing the paper’s title to ‘Transient agarose spot (TAS) assay: An improved method to investigate fibroblast migration’.
In abstract: “ …standard scratch assay which has numerous limitations. This article presents a detailed description of our transient agarose spot (TAS) assay, with instructions for its routine application. Advantages of combined use of different functional assays for fibroblast activation in drug development are also discussed …”
In introduction: “…an aggressive form of interstitial pneumonia characterised by excessive accumulation…”
“…primary disease can virtually affect any organ, making this a key determinant of health…”
“…fibroblasts comprise the main cellular component of scar tissue…”
“…composed of collagen and fibronectin [9,10]. These processes lead to excessive deposition of ECM which displaces native cells…”
“…clinical studies have confirmed multikinase inhibitors significantly slow progression of fibrosis by suppressing fibroblast activation…”
“Recently, several microplate-based assays have been described…” Note this refers to work published in 2009, so is this truly recent?
The above are offered as examples where minor reshaping of delivery could improve the readability of this strong technical effort.
Round 2
Reviewer 3 Report
I was convinced by the authors and other reviewers to accept the manuscript in the current form.
Reviewer 4 Report
All my comments have been taken into account and the text has been sufficiently corrected.